# Enhanced Antibacterial Performance of Chitosan/Corn Starch Films Containing TiO_2_/Graphene for Food Packaging

**DOI:** 10.3390/polym14183844

**Published:** 2022-09-14

**Authors:** Zhiyuan Liu, Xueqin Liu

**Affiliations:** 1Key Laboratory of Advanced Technologies of Materials, Ministry of Education, School of Materials Science and Engineering, Southwest Jiaotong University, Chengdu 610031, China; 2Henan Key Laboratory of Medical Polymer Technology and Application, Tuoren Medical Device Co., Ltd., Changyuan 453400, China

**Keywords:** chitosan, corn starch, nano-titanium dioxide, graphene

## Abstract

Chitosan (CTS)/corn starch (CSH)/nano-TiO_2_/graphene (Gr) antibacterial active packaging films were prepared by ultrasonic-assisted electrospray deposition and solution-casting methods, and the effects of the TiO_2_:Gr mass ratio and ultrasonication power on their morphology and mechanical, optical, thermal, barrier, and antibacterial properties were investigated. The film fabricated at a TiO_2_:Gr ratio of 6:4 and an ultrasonication power of 160 W exhibited a uniform distribution of the nanofillers in the CTS/CSH matrix and significantly enhanced the mechanical, barrier, and water-resistance properties. Furthermore, this film demonstrated superior ultraviolet and visible light-shielding characteristics as compared with those of the non-filled CTS/CSH film, while its *Escherichia coli* and *Staphylococcus aureus* inhibition efficiencies were equal to 96.67 ± 0.09% and 99.85 ± 0.13%, respectively. Therefore, the film can effectively prevent food spoilage, indicating its potential for food-packaging applications.

## 1. Introduction

The environmental problems posed by the widespread use of poorly biodegradable synthetic polymers as food-packaging materials necessitate the development of biodegradable polymers with high antimicrobial activity. Polysaccharide-based polymers are particularly suitable for this purpose, owing to their low costs, avirulence, and biodegradability [1]. Chitosan (CTS), one of the most popular biopolymers, is widely utilized in food packaging because of its good antifungal, antibacterial, and film-forming properties [2]. However, because CTS films exhibit poor mechanical characteristics and a high water solubility, pure CTS has limited practical applicability and, therefore, is often compounded with other materials. Corn starch (CSH) is an abundant natural polymer and a promising material for food-packaging and preservation applications because of its renewability, biodegradability, edibility, low cost, high mechanical resistance, and its ability to prevent moisture loss and undesired oxidation [3,4]. However, the simple utilization of polysaccharides for film preparation does not satisfy the market-related food preservation requirements and the performance of such films is usually very poor.

The polysaccharide-based polymers used as food-packaging materials are usually supplemented with nanoparticulate fillers such as nano-TiO_2_, which exhibits good photocatalytic/antibacterial properties and represents a safe food additive. TiO_2_ NPs were also found to be capable of ultraviolet (UV) light absorption and ensured the transparency of composite edible films [5]. Zhang et al. [6] prepared a CTS/TiO_2_ composite film with high antimicrobial activity against four typical foodborne pathogens (*E coli*, *S aureus*, *C. albicans*, and *A. niger*) under visible light irradiation. Apart from TiO_2_, graphene (Gr) is also commonly used as a nanofiller to improve the mechanical properties of films and endow them with antimicrobial activity [7,8]. Gonzalez et al. [9] showed that Gr antibacterial activity was influenced by the degree of Gr dispersion in polysaccharides. Although the aforementioned studies reported considerable enhancements of the film properties after the introduction of TiO_2_, Gr, CTS, or starch, the reproducibility of such studies was very poor, owing to the complexity of the chemical modification during film preparation. Ultrasonication enables the high-frequency agitation of aggregated NPs and film-forming compounds that significantly shortens their contact time and reduces friction forces [10]. In addition, ultrasonication power differences result in the continuous growth and collapse of cavitation bubbles in solution. Thus, after the application of a cyclic friction vector, cavitation leads to the uniform distribution of particles in the mixture [11]. Estrada-Monje et al. [12] investigated the ultrasonication-assisted dispersion of TiO_2_ NPs and concluded that the specified method produced a smaller number of agglomerates than mechanical stirring. Moreover, the TiO_2_ NPs added to an ethylene-co-vinyl acetate solution were evenly distributed in the composite film, which considerably increased its stability.

Many researchers demonstrated that ultrasonic treatment decreased the viscosity of starch dispersions and increased their solubility and transparency. However, none of the related research studies considered the effect of the nanoparticle content on the film performance at different ultrasonication powers. Moreover, neither CTS/CSH/TiO_2_/Gr nanocomposite films nor the effects of the TiO_2_ and Gr mass ratio on the film properties have been examined previously. In order to reduce the agglomeration of nanofillers, we prepared CTS/CSH/TiO_2_/Gr nanocomposite films by an ultrasonic-assisted electrospray deposition method to study the effect of their TiO_2_:Gr ratios on the film performance. The ultrasonic treatments of the film properties were evaluated by scanning electron microscopy (SEM), X-ray diffraction (XRD), differential scanning calorimetry (DSC), and thermodynamic measurements. These obtained results highlight the essential roles of two-component nanofillers and ultrasonic-assisted electrospray deposition in enhancing the performance of active packaging materials.

## 2. Materials and Methods

### 2.1. Materials

CTS (deacetylation degree: 88%, molecular weight (Mw): 165 kDa, medium viscosity, allergen-free, water insoluble, and soluble in acid media) was supplied by Sinopharm Chemical Reagent Co., Ltd. (No. 20120330, Shanghai, China). CSH (moisture content: 11.9%, amylose content: ~60%, and gelatinization temperature: 80 °C) was supplied by Sigma (St. Louis, MO, USA). Anatase TiO_2_ NPs (Mw: 79.87, 99.9% metals basis; particle size: 10 nm) were supplied by Nanoshel LLC (New York, NY, USA). Graphene (oxygen content: 0.5%, sulfur content: 0.5%, thickness: 3.4–8 nm, lamellar size: 5–50 μm, 6–10 layers, and purity > 95%) was supplied by Carboniferous Graphene Technology Co., Ltd. Glycerol and acetic acid (36 wt %) were obtained from Beijing Beihua Fine Chemicals Co., Ltd. (Beijing, China).

### 2.2. Preparation of CTS/CSH Films

First, CTS powder was dissolved in 1 wt % (such as 1 mL acetic acid in 99 mL distilled water) aqueous acetic acid to obtain a 2 wt % solution (such as 2 g CTS powder in 98 mL of 1 wt % acetic acid). The resulting mixture was supplemented with 30 wt % glycerol (CTS equivalent) for better plasticization and then stirred at room temperature (300 r/min, ~25 °C) with a magnetic stirrer (85–2, Changzhou Guohua Electric Co., Ltd., Changzhou, China) until its complete dissolution to obtain a CTS precursor solution. Next, a 6 wt % CSH suspension (such as 6 g CSH powder in 94 mL distilled water) was prepared using distilled water; then, it was thoroughly stirred and homogenized, fully gelatinized at 85 °C for 60 min, and cooled down to room temperature (~25 °C). The gelatinized body was then treated with glycerin to achieve a glycerin loading of 20 wt % (CSH equivalent) for better plasticization, after which magnetic stirring was performed for another 60 min to obtain a homogeneous aqueous dispersion [13]. The latter was subsequently ultrasonicated for 30 min to produce a CSH precursor solution. 

The CTS solution was mixed with the CSH precursor solution at a mass ratio of 1:1. After stirring, 90 g of the resulting mixture was uniformly and slowly poured onto a plastic plate (20 cm × 20 cm), slightly solidified, and dried in a vacuum oven at 45 °C for 6 h. The obtained film was further dried in air for 48 h until acetic acid was sufficiently volatilized. Except for the drying step, the entire process was performed at room temperature (~25 °C) and a relative humidity (RH) of <45%.

### 2.3. Preparation of CTS/CSH/TiO_2_/Gr Composite Films

To maximize the electrospray deposition efficiency, the spray solution normally contains low concentrations of TiO_2_ and Gr dissolved in ethanol. However, due to the aggregation of TiO_2_ and Gr NPs, 0.4 g of the TiO_2_-Gr mixture with variable TiO_2_:Gr ratios was added to ethanol (5 mL), and the obtained suspension was ultrasonicated by an XH-2008D ultrasonic instrument (Xianghu Development Co., Ltd., Beijing, China) equipped with a reactor containing a thermostatic water bath (temperature accuracy: ±1 °C), a mechanical stirrer, and a microtip probe (diameter: 8 mm). The operating frequency of the ultrasonicator was 25 kHz, the sonication time was 30 min, and ultrasonication power varied between 0, 80, 120, 160, and 200 W to ensure high stability of the electrospray solutions with different concentrations. The (TiO_2_ + Gr):(CTS + CSH) mass ratio was set to 1:20, while the TiO_2_:Gr mass ration varied between 10:0, 9:1, 8:2, 7:3, 6:4, and 5:5. Subsequently, the suspension was filtered through a 5 mL syringe with a single-needle configuration (outside diameter: 200 μm; inside diameter: 100 μm) and continuously pushed by a syringe pump (Zhejiang University Medical Instrument, Hangzhou, China) at a rate of 0.25 mL/h, using an electro-spraying apparatus equipped with an 8 kV power supply (Tianjing High-Voltage Power Supply Co., Ltd., Tianjing, China). The spraying distances between the syringe nozzle and the CTS/CSH films were set to 5 cm. The CTS/CSH/TiO_2_/Gr composite films were vacuum-dried at room temperature for 24 h to completely remove all solvent residues prior to the further use.

### 2.4. Dispersion Stability Testing

The CTS/CSH/TiO_2_/Gr solutions were ultrasonicated for 30 min at ultrasonic powers of 0, 80, 120, 160, and 200 W. After 24 h, all solutions were photographed and the nanofiller dispersions were evaluated by observing their settlement. To access the dispersion stability more accurately, zeta potentials of the prepared composite solutions were measured at 25  ±  0.1 °C using a Nano-ZS laser particle analyzer (Zetasizer Nano ZS90, Malvern Co., Ltd., Worcestershire County, UK).

### 2.5. Scanning Electron Microscopy (SEM) Observations

SEM (Su 66000, Hitachi, Japan; accelerating voltage: 20 kV) was used to observe the morphologies of the pure CTS/CSH and filler-containing films. Prior to analysis, the samples were cut to appropriate sizes and coated with thin gold layers under vacuum. The film surfaces were examined at a magnification of 10,000× and a magnification of 100,000× for the nanoparticle distributions.

### 2.6. Fourier Transform Infrared (FTIR) Spectroscopy Studies

Films with a TiO_2_:Gr ratio of 6:4 were prepared at the same ultrasonication power (160 W) but different TiO_2_:Gr ratios at 23 °C and 62.3% RH. For each sample, FTIR (Suzhou Leiden Scientific Instrument Co., Ltd., Suzhou, China) spectra were recorded at three random locations in the range of 650-4000 cm^−1^ (32 scans; resolution: 4 cm^−1^).

### 2.7. XRD Analysis

XRD patterns of the composite films were recorded on a Bruker AXS D8 Advance X-ray diffractometer (Bruker AXS GmbH, Karlsruhe, Germany; Ni-filtered Cu *K*_α_ radiation) in the 2*θ* range of 5–80° at a current of 40 mA, voltage of 40 kV, step size of 0.02°, and counting time of 1 s. The wavelengths of X-rays are in the range from 0.01 nm to 10 nm, which corresponds to energies in the range from 0.125 to 125 keV.

### 2.8. DSC Analysis

Films’ thermal properties were evaluated by DSC (Perkin Elmer, Boston, MA, USA). Samples with masses of approximately 10 mg were weighed in a hermetic pan to avoid water losses and were heated from 10 to 350 °C at a rate of 10 °C/min under nitrogen atmosphere. An empty hermetic pan was used as a reference. The experiment was performed in triplicate, and the best dataset was selected for analysis by the Origin 2017 software to locate the endothermic peaks of water evaporation (*Tw*) and the melting point (*Tm*) and to determine the enthalpy of melting (∆*Hm*).

### 2.9. Density measurement

Each film was cut into specimens of the same size, and its thickness (*d*) was measured with a thickness gauge (Mitutoyo Absolute, Tester Sangyo Co., Ltd., Tokyo, Japan) at five locations in the sample center and at four locations around the center. The average value of the obtained thicknesses was calculated. The estimated accuracy of the median was 1 μm. Film mass (*m*) was measured by an analytical scale with an accuracy of 0.1 mg, and the sample area (*s*) was also calculated and recorded. Finally, film density (*ρ*) was approximately calculated using Equation (1). Density measurements were repeated five times for each film sample, and the obtained results were presented as the corresponding means.
(1)$$\rho =\frac{m}{s\cdot d}$$

### 2.10. WVP Measurements

The CTS/CSH, CTS/CSH/TiO_2_, and CTS/CSH/TiO_2_/Gr films were cut into round specimens with a radius of 33 mm and tested by a water vapor permeameter (PERME W3/031, Lab Think Instruments Co., Ltd., Jinan, China) in accordance with the ASTM E96/E96M standard method [14]. Each sample was analyzed thrice.

### 2.11. Mechanical Properties

Mechanical properties of the films were evaluated using a texture analyzer (TA XT plus 50, Stable Micro Systems Ltd., Vienna, UK) [15]. The films were cut into uniform 60 mm × 10 mm samples and tested at a tensile speed of 250.00 mm/min to determine their TS, EAB, and Young’s modulus (E) values at room temperature [16].

### 2.12. Water Contact Angle (WCA), Moisture Content (MC), and Water Solubility (WS) Studies

The WCA for the CTS/CSH and various CTS/CSH/TiO_2_/Gr composite films were determined to evaluate their wettability using a static water contact angle instrument (POWEREACSH JC2000D3, Shanghai, China).

MC was determined by measuring the weight loss upon drying to a constant weight in an oven at 105 °C as follows:(2)$$\mathrm{MC}\;\left(\%\right)=\frac{M_{w}-M_{d}}{M_{w}}\times 100,$$
where *M*_w_ and *M*_d_ are the films’ initial weight and dry weight, respectively.

The initial dry matter content was determined by drying to a constant weight in an oven at 105 °C (*W*_i_), after which each film was immersed into distilled water (50 mL) at 25 °C. After 24 h of incubation, the samples (2 cm × 2 cm) were dripped and dried to a constant weight at 105 °C to determine the weight of the dry matter not solubilized in water (*W*_f_). WS was calculated as:(3)$$\mathrm{WS}\;\left(\%\right)=\frac{W_{i}-W_{f}}{W_{i}}.$$

### 2.13. Light Transmittance and Surface Color

UV-visible transmission spectra were recorded in the wavelength range of 200–800 nm on a UV-vis spectrophotometer (UV-1800 (PC), Shanghai AuCy Instrument Co., Ltd., Shanghai, China). Surface color was determined by a chromometer (Konica Minolta, CR-400, Tokyo, Japan) using a white plate as the standard background (*L* = 85.52, *a* = 3.05, *b* = 0.17). A five-point sampling method was applied to select five points on each sample surface and obtain an average measurement value. The total color difference (Δ*E*) was calculated as
(4)$$\Delta E=\sqrt{{\left(\Delta L\right)}^{2}+{\left(\Delta a\right)}^{2}+{\left(\Delta b\right)}^{2}}$$
where Δ*L*, Δ*a*, and Δ*b* are the differences between the color values of the standard color plate and those of the film sample.

### 2.14. Oxygen Resistance

Oxygen resistance was determined by sodium thiosulfate titration. A clean test tube was filled with fresh peanut oil (10 g), sealed with a certain film sample (2 cm × 2 cm), and stored in an incubator at 60 °C for 10 d. The peroxide value (PV, g/100 g) of the peanut oil was determined by sodium thiosulfate titration according to the formula
(5)$$PV=\frac{\left(V-V_{0}\right)\times C\times 0.1269}{M}\times 100\;$$
where *V* is the volume of the sodium thiosulfate standard solution consumed by the sample (mL), *V*_0_ is the volume of the sodium thiosulfate standard solution consumed by the blank (mL), *C* is the concentration of the sodium thiosulfate standard solution (M), 0.1269 is the amount of iodine (g) corresponding to 1 mL of the sodium thiosulfate solution, and *M* is the mass of the titrated oil (g).

### 2.15. Antibacterial Properties

Films were cut into 20 mm × 20 mm rectangles each weighing ~100 mg and sterilized on a super-clean bench under UV irradiation for 2 h. *E. coli* and *S. aureus* were inoculated in LB liquid medium (50 mL) and vibrated at 37 °C and 100 rpm for 24 h to expand the strain culture. Subsequently, a 1 mL aliquot of each bacterial suspension was added to freshly prepared NA liquid medium (50 mL), and the suspension concentration was adjusted to 1–2 × 10^6^ colony-forming units (CFU) mL−1 and supplemented with a membrane sample (1 g). The membrane samples were sterilized by ultraviolet radiation for 30 min before addition. Then, we plated them on nutrient agar, which was incubated at 37 °C for 24 h. CSH was the control. The colonies were then counted. The antibacterial efficiency (*B_R_*) was calculated as follows:*B_R_* = (*B* − *C*)/*B* × 100%(6)
where *B_R_* is the bacteriostatic rate (%), *B* is the average colony density (CFU g^−1^) for the blank control, and *C* is the average colony density (CFU g^−1^) for the nanofibers.

### 2.16. Statistical Analysis

The result was expressed as the mean ± standard deviation. The SPSS 24.0 statistical analysis system was used for analysis of variance (ANOVA) and Duncan’s multi-range teats were used for determining significant differences from the other groups (*p* < 0.05).

## 3. Results and Discussion

### 3.1. Dispersibility of Nanofillers in Film Precursor Solutions

Figure 1a,b display the SEM and TEM images of Gr powder, respectively, indicating that Gr has a lamellar structure with thin, wrinkled layers. Figure 1c,d depict the TiO_2_ NPs powder obtained by transmission electron microscopy (TEM, recorded at an acceleration voltage of 200 kV, Tecnai G2 F20 S-Twin, Thermo Fisher Scientific Co., Ltd., Minneapolis, USA), which show the columnar particles with an average size of approximately 10.76 ± 1.73 nm.

These results confirm the good dispersibility of TiO_2_ NPs in the CTS/CSH matrix, while the zeta potential value of the precursor solution obtained at an ultrasonication power of 160 W was −14.33 ± 0.45 mV (Table 1). However, at a TiO_2_:Gr ratio of 9:1, the dispersion stability significantly decreased, and the precursor solution separated into three different layers: a clear upper layer, a lightly colored middle layer corresponding to a uniform CTS/CSH/TiO_2_/Gr dispersion, and a deeper colored lower layer containing precipitated TiO_2_/Gr particles. The relatively small amount of added Gr limits its expansion. This phenomenon caused a significant drop in the solution zeta potential to −10.31 ± 0.13 mV and decreased the stability of the system. Upon increasing the Gr content, the upper layer contained a very small amount of a clear liquid after 24 h of storage, while the black (due to the high Gr content) lower layer essentially consisted of the CTS/CSH/TiO_2_/Gr dispersion. The results revealed that the stability of the precursor solution toward a separation increased with an increasing Gr content, which was accompanied by the increase in its zeta potential values [17]. This behavior was attributed to the ability of Gr to undergo expansion in aqueous solutions and the wrapping of the TiO_2_ NPs, thus increasing their dispersibility. Furthermore, the ultrasonic treatment strongly influenced the dispersion stability of the TiO_2_ NP solution with TiO_2_:Gr = 6:4. The zeta potential value of this solution without the ultrasonic treatment was only −8.40 ± 0.22 mV. As the ultrasonic power increased, the zeta potential value (absolute value) continuously increased to −19.76 ± 2.28 mV. At high temperatures and pressures, the strong shock waves generated by the ultrasonic cavitation promoted the decomposition of the nanofiller aggregates into smaller aggregates, which were more uniformly dispersed in the system [18].

### 3.2. Morphology of Composite Films

The surface microstructures of the composite films fabricated at an ultrasonication power of 160 W were examined by SEM (Figure 1e). The CTS/CSH film was characterized by a continuous surface without any particles or pores, and no phase separation was observed between the two constituent polymers, which suggested that the CTS, CSH, and glycerin (plasticizer) phases were very compatible with each other. The morphology of the film with a TiO_2_:Gr ratio of 10:0 demonstrated that excess TiO_2_ facilitated the formation of uneven particles, which was likely caused by the aggregation of the TiO_2_ NPs that were prone to aggregation, in accordance with the conclusion of Li et al. [19]. The Gr-containing film had a rough (wrinkled and corrugated) texture primarily due to the stacking of Gr sheets. At a TiO_2_:Gr ratio of 6:4, the Gr’s surface was densely packed with TiO_2_ NPs and also contained slightly agglomerated TiO_2_ NPs, which suggested relatively strong bonding between the Gr and TiO_2_ particles. Thus, the formation of a dense and continuous layered film structure was attributed to the predominance of ionic bonding between the film components [20].

### 3.3. FTIR Studies

Interactions between the main components are important for films [21]. In the FTIR spectrum of TiO_2_, the wide band at 650–1000 cm^−1^ corresponds to the Ti–O–Ti stretching vibrations, while the peak at ~1100 cm^−1^ is ascribed to Ti–OH stretching (Figure 2a). The characteristic bands at ~1386 and 1637 cm^−1^ reveal the presence of O–O moieties and water, respectively, while the peak centered at 3436 cm^−1^ is attributed to the presence of moisture and hydrogen-bonded OH groups on the TiO_2_ surface. In the FTIR spectrum of Gr, the bands at 1560 and 1630 cm^−1^ are attributed to the stretching of C–C and C=C bonds, and the peak at 3440 cm^−1^ is ascribed to the stretching vibration of –OH groups originating from either externally adsorbed water molecules or the incomplete reduction of -OH groups after the thermal exfoliation of expandable Gr [22]. The CTS/CSH spectrum features a typical amide I (C=O) band at 1635 cm^−1^ and an amide II (C–N) band at 1598 cm^−1^. The fundamental stretching modes of water and carbohydrate OH groups are usually observed in the region of 3100–3600 cm^−1^, as indicated by the O–H tensile vibrations at 3295 cm^−1^ in the spectra of the CTS/CSH and CTS/CSH/TiO_2_ films. Compared with the CTS/CSH/TiO_2_ film, the CTS/CSH/TiO_2_/Gr film exhibits a much stronger C–O–C peak at 1000 cm^−1^ and a red-shifted –COO^−^ peak at 1654 cm^−1^, which indicates the presence of the Gr particles embedded into the CTS/CSH matrix. Therefore, it was concluded that strong ionic and hydrogen bonds were present in the films containing both fillers. The intensity of the –OH peak increased with the increasing Gr content (except for the 5:5 sample) mainly because of the presence of a large number of hydroxyl and methylene groups on the Gr’s surface and edges [22]. The influence of ultrasonication on the FTIR spectra of the composite films was attributed to the cavitation-induced destruction of hydrogen bonds between the CTS/CSH matrix and Gr or TiO_2_ particles, which released free –OH groups and increased the intensity of the corresponding peak [16]. Meanwhile, the number of hydrophilic components was considerably reduced due to the release of –OH groups. Hence, the observed increase in the film hydrophobicity was consistent with the conclusions of Abral et al. [23].

### 3.4. XRD Analysis

The XRD pattern of TiO_2_ exhibits strong diffraction peaks at *2θ* = 25.52°, 37.96°, 48.24°, 54.02°, 62.84°, 69.00°, 70.48°, and 75.28°, corresponding to the (101), (103), (101), (004), (200), (105), (211), (204), (116), (200), and (215) crystallographic planes of the anatase TiO_2_, respectively (Figure 3a). The XRD of Gr shows an intense peak at 2*θ* = 26.48° due to the reflection from the (002) plane of the graphitic carbon structure with a spacing of 0.34 nm, suggesting the existence of π–π stacking between the Gr sheets [24].

The peak at 2*θ* = 54.62° corresponds to the Gr (004) plane. The XRD pattern of the CTS/CSH film contains a broad peak and a distinct amorphous scattering background characteristic of semi-crystalline polymers. The main peaks at 20.88° and 35.72° indicate the high crystallinity of the CTS/CSH matrix, while the formation of intermolecular hydrogen bonds between the CTS and CSH particles restricts the movement of molecular chains and thus inhibits the crystallization process [25]. The obtained results are in good agreement with those of Ren et al. [4]. The XRD pattern of the CTS/CSH/TiO_2_ film exhibits the characteristic peaks of crystalline TiO_2_ at 2*θ* = 25.20°, 37.78°, 47.86°, 53.68°, 54.94°, and 62.52°, indicating the successful incorporation of the TiO_2_ NPs into the CTS/CSH matrix. Interestingly, upon the addition of TiO_2_, the original CTS/CSH peaks at 25.52° and 20.88° shifted to 25.20° and 19.78°, respectively, indicating a concomitant increase in the number of defects in the CTS/CSH/TiO_2_ film structure. This phenomenon was attributed to the partial destruction of the CSH crystals due to gelatinization and the effects of the strong hydrogen bonding between the CTS, TiO_2_, and CSH components on the crystal structure of the composite film and its properties [26]. The pattern of the CTS/CSH/TiO_2_/Gr film contains the characteristic peaks of Gr (at 26.84° and 54.90°) and TiO_2_ (at 25.66°, 38.08°, and 48.28°), confirming that both nanofillers were successfully incorporated into the CTS/CSH matrix.

Compared with the CTS/CSH/TiO_2_ film, the characteristic peak intensity of TiO_2_ in the CTS/CSH/TiO_2_/Gr composite film was relatively small due to the low concentration of the TiO_2_ NPs in the quaternary composite film and the ability of Gr to form a composite structure with TiO_2_. It is noteworthy that the Gr peak at 26.48° obtained for the ternary composite film shifted to 26.84° for the quaternary composite film, suggesting a decrease in the Gr interlayer distance. This phenomenon was attributed to the poor dispersibility of the Gr particles due to their relatively high content, which restricted their movement and caused the entanglement of CTS/CSH polymer chains in accordance with the previous findings [26].

### 3.5. DSC Analysis

The DSC thermal spectra of the CTS/CSH, CTS/CSH/TiO_2_, and CTS/CSH/TiO_2_/Gr films are shown in Figure 3b. The first heat absorption peak of the composite films located in the region of 69.45–153.44 °C is attributed to the evaporation of absorbed water, bound water, and glycerin plasticizer [27,28]. The endothermic peak of the CTS/CSH film is located at 118.54 °C, while for the CTS/CSH/TiO_2_ film its position is shifted to 97.46 °C due to the high concentration of TiO_2_ NPs that facilitated their agglomeration and produced a relatively loose structure. The endothermic peak of the CTS/CSH/TiO_2_/Gr film (124.52 °C) was higher than that of the CTS/CSH film, owing to the higher dispersibility of the TiO_2_ NPs and Gr, the good compatibility of these species with the CTS/CSH matrix, and the strong interfacial interactions between the film components, making the material structure more compact and thermally stable [29]. The endothermic peak in the temperature range of 247.96–297.69 °C was mainly attributed to the chemical degradation and deacetylation of functional groups (Table 2), such as the amine group and the hydroxyl group [30]. In addition, the film-melting enthalpy (∆*H_m_*) increased after the addition of the nanofillers, possibly because of strong interactions between the nanofillers and the CTS/CSH matrix. The following interactions may exist in the composite systems: (i) intermolecular interactions between CTS and CSH; (ii) hydrogen bonding between CTS/CSH and the plasticizer (glycerin); (iii) interactions between CTS and the swollen CSH particles; (iv) interactions between CSH and the swollen CSH particles; (v) interactions between swollen CSH particles and TiO_2_; (vi) interactions between the swollen CSH particles and Gr. Therefore, both the TiO_2_ and Gr could adsorb swollen CSH particles in the film precursor solution to hinder the surface interaction of these particles with CTS/CSH and thereby enhance the interaction between the polymer macromolecules contained in the continuous phase and increase its melting enthalpy [31]. The outlined processes can explain the observed increase in the intensity of the corresponding FTIR absorption peak.

### 3.6. WVP

Figure 3c shows that all the films exhibit good water vapor barrier properties. The incorporation of TiO_2_ NPs usually decreases the WVP of biopolymers; however, this effect occurs in a limited loading range, while very high and low filler loadings increase the WVP value. The WVP of the films prepared in this work first decreased and then increased with the increasing TiO_2_:Gr ratio. Its lowest value was obtained at a ratio of 6:4 (2.74 ± 0.10 × 10^−12^ g cm^−1^ s^−1^ Pa^−1^), which was 23.25% smaller than the WVP of the CTS/CSH film (3.57 ± 0.23 × 10^−12^ g cm^−1^ s^−1^ Pa^−1^). This behavior was attributed to the following reasons. (i) Gr is strongly hydrophobic and could therefore block the passage of water molecules. (ii) The addition of TiO_2_ and Gr particles inhibited the interactions of free hydroxyl groups with water and reduced the utilization of hydrophilic groups to suppress the passage of water through the film [32]. (iii) The results of the SEM analysis revealed that the two nanofillers were well dispersed in the CTS/CSH matrix at the specified loading, and the combination of these highly dispersed NPs and the matrix produced a more tortuous path for water vapor molecules [33]. (iv) The addition of the nanofillers induced the rearrangement of the CTS/CSH polymer chains so that they were arranged in a more orderly manner, reducing the free volume of the polymer and thereby its WVP [26]. (v) The thickness of the film might also influence this process.

The amount of added TiO_2_ NPs is apparently correlated with ultrasonication power. For example, at 0 W, the WVP of the film with a TiO_2_:Gr ratio of 6:4 is lower than that of the film with a TiO_2_:Gr ratio of 7:3; however, at 80 W, there is a contrast between the two films. At 120 W, their WVP values are the same, and at 160 W, the WVP of the 6:4 film is much lower than that of the 7:3 film. Hence, the ultrasonication power of 160 W is most suitable for the 6:4 ratio. After the ultrasonication at a power of 160 W, the WVPs of the CTS/CSH film and the composite film with a TiO_2_:Gr ratio of 6:4 decreased to 3.04 ± 0.21 × 10^−12^ (by 14.85%) and 2.02 ± 0.06 × 10^−12^ g cm^−1^ s^−1^ Pa^−1^ (by 26.68%), respectively. When a certain ratio between the added filler amount and ultrasonication power is exceeded, the water barrier properties of the composite film deteriorate. In addition, compared to the composite film containing only TiO_2_ NPs, the addition of mixed NPs can reduce the WVP of the composite film to a certain extent. It is possible that the cavitation-induced turbulence caused high-speed oscillations and particle collisions, which promoted the interactions between various film components and facilitated their uniform dispersion [34]. Owing to the high-pressure vibrations of ultrasound, the polymer possessed a higher kinetic energy, which promoted the interactions between the TiO_2_ and Gr particles, making them more closely aligned and thereby reducing the WVP value. However, at an extremely high ultrasonication power, the polymer may become damaged, which causes the penetration of water molecules and increases its WVP [16]. Therefore, the addition of TiO_2_ and Gr particles decreased the WVP more effectively than the addition of TiO_2_ NPs alone.

### 3.7. Thickness and Density

Within an appropriate power range (80–160 W), ultrasonication creates regions with increased pressure inside the liquid to facilitate the formation of a denser and more uniform polymer network structure (Figure 4a,b). The lower film thickness and higher polymer density resulting from the restructuring of the four film components caused the absorption of ultrasonic kinetic energy that produced more active short chains and filled pores. However, an extremely high ultrasonication power (200 W) could damage the polymer and generate additional holes in its surface to reduce the material density, as was observed previously [16].

The thicknesses of all films were between 0.076 ± 0.001 and 0.105 ± 0.003 mm, and the densities varied between 0.14 ± 0.00 and 0.15 ± 0.01 g/cm^3^. Moreover, the composite films were only insignificantly (*p >* 0.05) denser than the CTS/CSH film. The thickness increase observed for the composite films likely originated from nanofiller addition. Roy & Rhim [35] reported the same rule. The strong interactions between the nanofillers and the CTS/CSH allowed the formation of a denser and more uniform composite structure and thereby increase their density. In addition, the oxygenated functional groups on the Gr’s surface coordinated with the bivalent body to enhance the performance of Gr complexes. Therefore, the metal ions formed denser aggregates and thus increased the film density [17]. With increasing Gr content, the density of the composite film decreased due to the low density of the Gr phase and its multi-layer sheet structure. Meanwhile, when Gr particles were embedded into the polymer matrix, their sheet layers were easily unfolded by the polymer to increase the thickness of the composite film and reduce its density.

### 3.8. Mechanical Properties

With decreasing the TiO_2_:Gr ratio, the TS and E values of the produced films first increased and then decreased, while an opposite trend was observed for the EAB. The maximal TS and E values of 20.01 ± 2.27 MPa and 366.42 ± 6.59 MPa, obtained at a TiO_2_:Gr ratio of 6:4 were 109.75% and 135.25% higher than those of the pure CTS/CSH film (TS: 9.54 ± 2.00 MPa, E: 155.76 ± 1.75 MPa), respectively. Meanwhile, the minimal EAB of 23.75 ± 4.04% obtained at the same filler ratio, was 47.08% lower than that of the pure CTS/CSH film (44.88 ± 3.06%) (Figure 5a–c). At low concentrations, TiO_2_ NPs were more likely to strongly interact with the CTS/CSH matrix, and the flow and deformation of macromolecular segments were limited by the strong interactions between either the nanofillers and the matrix or between nanofiller particles, which increased the film rigidity and reduced its EAB [36]. However, as the Gr content continuously increased, TS decreased to 22.44 ± 6.04 MPa.

Ultrasonication also affected the mechanical properties of the composite films. Thus, the treatment at a power of 160 W increased EAB, E, and TS, while the deterioration of the mechanical properties was observed at a larger power of 200 W. At the same ultrasonication power, the mechanical properties were first enhanced and then deteriorated with an increasing Gr content. This behavior was attributed to the effect of the applied shear stress, which broke the weak intermolecular hydrogen bonds between the different components and caused a disordered arrangement of polymer chains to promote the sliding of the crystal surfaces [37]. Under normal conditions, an extremely high Gr loading may obscure the centers of action where other molecules gather and thereby hinder molecular interactions and promote the formation of free particles in the film structure, thus decreasing its stability and TS and EAB magnitudes. Notably, although ultrasonication deteriorated the mechanical properties of the composite films, they still satisfied the food-packaging safety requirements.

### 3.9. Water Contact Angle (WCA), MC, and WS

The WCA of the CTS/CSH film was 70.90 ± 1.56°, indicating its good hydrophilicity, which was mainly due to the large number of hydrophilic groups in the CSH particles, and the -NH_2_ and -COOH groups in the CTS particles with a high affinity for water molecules. The WCA value of CTS/CSH/TiO_2_ decreased to 66.69 ± 1.35°, owing to the high hydrophilicity of the TiO_2_ NPs [6]. With the increase in the Gr content, the WCA of the composite films gradually increased, and its maximum value of 89.78 ± 1.46° was achieved at TiO_2_:Gr = 6:4, indicating that the hydrophobicity of the composite film increased, mainly because the uniform dispersion of hydrophobic Gr particles in the composite structure reduced its degree of hydrophilicity [38]. However, when the Gr content was further increased, the WCA value slightly decreased due to Gr aggregation, which reduced the effective Gr’s surface area, thereby increasing the number of hydrophilic groups on the film’s surface [39].

The water-binding capacity of packaging materials significantly affects their mechanical and barrier properties. Therefore, the WS and swelling rate of biopolymer composites are important parameters for characterizing their practical applicability. In the composite films, TiO_2_ behaved as typical inorganic NPs containing hydrophilic (e.g., hydroxyl) surface groups (Figure 5d,e). When TiO_2_ was evenly dispersed in the polymer matrix, the polarity and hydrophilicity of the composite film increased with the increasing TiO_2_ content. However, agglomeration at an overly large TiO_2_ content decreased the hydrophilic surface area and hence the polarity and hydrophilicity of the composite film [16]. The lowest MC was observed for the pure CTS/CSH film (17.75 ± 0.85%), while the highest MC was observed for the CTS/CSH/TiO_2_ film (21.92 ± 0.24%), as the latter film featured the highest TiO_2_ content. The presence of hydrophilic TiO_2_ NPs allowed more water molecules to be preserved in the film and thereby increased the MC [16]. Notably, the MC decreased with the increasing Gr loading, possibly because at high Gr loadings, the carbon atoms on the Gr’s surface were tightly bound and had no defect range, which hindered the passage of water molecules. The good dispersion of Gr in the matrix and the distortion caused by the height–length–width ratio of Gr lengthened the path of water molecules and reduced the MC.

Figure 5f shows that the WS decreased after the introduction of TiO_2_ and Gr. Specifically, the WS decreased with the increasing Gr content. The lowest WS of 8.78 ± 0.46% was observed at a TiO_2_:Gr ratio of 6:4 and was 36.83% lower than that of the pure CTS/CSH film (13.90 ± 0.56%), indicating that the water resistance increased upon the nanofillers’ incorporation. This phenomenon was attributed to the presence of hydrophobic acetyl groups and amino groups in CTS. The former groups could bind to the Gr’s surface, while the latter improved the dispersibility of Gr in water, which resulted in a uniform distribution of Gr in the composite film and, hence, in a reduced film hydrophilicity [38]. In addition, hydrogen bonds could be formed between TiO_2_ and the CTS/CSH matrix to reduce the films’ WS. Similar results were reported by Divsalar et al. [40], who showed that the NPs incorporated into water-soluble CTS-based films effectively cross-linked the polymer network to form a tighter structure with a lower film-swelling index.

With the increasing ultrasonication power, the WAC value first increased and then decreased, which was mainly due to the more uniform distribution of the nanofillers that can be more evenly distributed in the matrix under high-intensity ultrasonic conditions, which increased their effective surface area. Moreover, the improvement of the surface roughness of the composite film furthered its hydrophobicity [39]. The MC value also first increased and then decreased with the increasing ultrasonication power. The initial increase was ascribed to the increased effectiveness of water molecules’ entry into the nanofiller within an appropriate ultrasonication power range due to the effects of the kinetic energy and thermal energy generated by ultrasonication. The WS of the composite films first decreased and then increased with the increasing ultrasonication power, although the corresponding difference was not significant (*p* > 0.05). This behavior was ascribed to the promotional effects of ultrasonic cavitation and kinetic energy supply on the NP dispersion, which allowed for a close combination of TiO_2_ NPs/Gr and CTS/CSH to increase the strength of intermolecular interactions and thus reduce the WS [41]. However, as the ultrasonication power was increased further, the corresponding thermal effect enhanced the movement of molecules and reduced the viscosity of the CTS/CSH matrix to increase the WS of the composite films and decrease their WAC and MC values.

### 3.10. Surface Color Measurements

Table 3 shows the colors of the pure CTS/CSH, CTS/CSH/TiO_2_, and CTS/CSH/TiO_2_/Gr films, indicating that the introduction of the nanofillers resulted in significant color changes (*p* < 0.05). The non-ultrasonicated CTS/CSH film was transparent, and its *L*, *a*, and *b* values were equal to 86.54 ± 0.03, 3.34 ± 0.00, and 2.11 ± 0.12, respectively. The TiO_2_-containing composite film was white and its *L* value (87.78 ± 0.04) exceeded that of the pure CTS/CSH film, i.e., the former film was whiter, which was consistent with the well-known whitening effect of TiO_2_ [42]. The introduction of the inherently black Gr resulted in significant film darkening and a decrease in its *L* value (*p* < 0.05) as compared with that of the CTS/CSH film. The lowest *L* of 30.79 ± 0.01 (and hence, the darkest color) was observed at the highest Gr content (TiO_2_:Gr = 5:5) and was 64.42% lower than that of the CTS/CSH film. After adding the nanofillers, the *L* value of the composite film was lower than that of the pure CTS/CSH film (*p* < 0.05), and the color change became more visible with an increase in the Gr content. The value of b first increased and then decreased; its largest magnitude (5.41 ± 0.31) was obtained for the CTS/CSH/TiO_2_ film, owing to the blending of the CTS and TiO_2_ NPs. Moreover, b decreased with the increasing Gr content because of the gradual accumulation of this filler. The addition of the nanofillers also resulted in a significant increase in ∆E (*p* < 0.05). The highest ∆E of 45.86 ± 0.01 was observed at a TiO_2_:Gr ratio of 5:5, which was attributed to the synergistic effect of the CTS, CSH, TiO_2_, and Gr components. Moreover, the films with TiO_2_:Gr ratios of 7:3, 6:4, and 5:5 exhibited similar colors because of the major influence of the black-colored Gr phase.

At the optimal proper ultrasonication power, the TiO_2_ and Gr particles can be evenly dispersed in the film matrix to obtain uniformly colored films. However, at a high power of 200 W, a larger number of bubbles were generated around the ultrasound probe. These bubbles gathered at the nodes of the acoustic field to protect the remaining solution from the effects of ultrasonic vibration. This phenomenon (called an over-processing effect) may promote the aggregation and agglomeration of Gr and TiO_2_, thus affecting the film color [16]. At present, the content of a particular component in the film structure is considered the main cause of the color difference, while the ultrasound treatment represents a secondary cause, which is in good agreement with the others results [43]. Specifically, the film color was mainly influenced by the TiO_2_:Gr ratio, whereas the color dependence on the ultrasonication-power-affected biopolymer size was found to be less important.

### 3.11. Visible and UV Transmittances

The UV and visible light transmittances of the ultrasonicated composite films were lower than those of the non-ultrasonicated pure CTS/CSH film (*p <* 0.05) (Table 4), indicating that the addition of TiO_2_ and Gr particles can significantly increase the films’ resistance to UV and visible light. The transmittances of the pure CTS/CSH film in the UV region at 280, 300, and 400 nm decreased to 1.32 ± 0.00% (by 98.25%), 4.24 ± 0.11% (by 95.07%), and 0.36 ± 0.01% (by 99.54%), respectively, upon the introduction of the TiO_2_ NPs. This behavior was attributed to the known UV-shielding effect of TiO_2_ due to the large bandgap energy (>2.8 eV) of this semiconductor [15]. The introduction of the Gr particles into the CTS/CSH/TiO_2_ film gradually increased its transmittance in the UV region, which was likely caused by the decrease in the relative content of the TiO_2_ NPs with better UV light-absorption properties. However, the light transmittance of the CTS/CSH/TiO_2_/Gr films remained much lower than that of the pure CTS/CSH film (*p <* 0.05). The latter exhibited a high transmittance in the visible light region with *T*_600_ and *T*_800_ values of 81.60 ± 0.70% and 82.27 ± 0.58%, respectively, which significantly decreased to (0.71 ± 0.11) and (3.15 ± 0.32)% after the addition of the nanofiller (*p <* 0.05). Moreover, the visible light transmittance continuously decreased with the increasing Gr content. Since Gr has a multi-layered sheet structure, it generates strong intermolecular forces between the Gr particles, which facilitates their dispersion in the CTS/CSH matrix and increases the roughness of the composite film, resulting in decreased light transmission [20]. Li et al. [38] found that after Gr’s addition, the light transmittance of a poly(lactic acid)/poly(butylene adipate-*co*-terephthalate) composite film decreased over the entire 350–800 nm region, confirming that Gr incorporation can effectively reduce the transmittance of composite films. The high specific surface areas of the TiO_2_ NPs and Gr particles facilitate the uniform distribution of these nanofillers in the CTS/CSH matrix, which increases their surface area and UV absorption efficiency, owing to the effective absorption of kinetic energy induced by ultrasonic cavitation [38]. However, the high ultrasonication power also increases the visible light transmittance. In summary, the light transmittance and anti-UV light performance of the composite film increase with an increase in the Gr content. This property of the studied films can help protect wrapped food and inhibit the oxidation of oil and other processes.

### 3.12. Oxygen Resistance

To protect foods that can be easily oxidized, the utilized packaging materials must possess a high oxygen barrier rate. The oxygen resistance of the CTS/CSH film significantly increased (*p* < 0.05) after the addition of the nanofillers (Table 5). The synergy between the CTS, CSH, and TiO_2_ components likely resulted in the formation of strong hydrogen bonds between various molecules of the precursor solution; it also decreased the free volume of the blended film system to inhibit the dissolution and diffusion of gases and thus increase the oxygen resistance in accordance with the findings of Tenore et al. [44]. Upon the introduction of the Gr particles, the oxygen resistance first increased and then decreased, with a minimum PV of 16.56 ± 0.04 g/100 g achieved at a TiO_2_:Gr ratio of 6:4. This behavior was mainly attributed to the effective blockage of the existing gas passages by the lamellar Gr structure [45]. In addition, at a TiO_2_:Gr ratio of 5:5, the oxygen transmission increased with the increasing Gr content because of the uneven distribution of the Gr particles. The oxygen resistance of the composite films was significantly higher than that of the PE film (22.99 ± 0.24 g/100 g), suggesting the wide range of their potential applications. Ultrasonication reduced the PV to a certain extent (*p >* 0.05), and the lowest PV value (16.48 ± 0.13 g/100 g) was achieved at an ultrasonication power of 160 W. This effect was ascribed to the depolymerization of the nanofillers under the high pressure associated with the rupture of cavitation bubbles and the intense heating of the residual bubbles combined with a violent mechanical action. Under these conditions, the agglomerated TiO_2_ NPs were decomposed into smaller particles to form a dense polymer structure containing Gr particles, which blocked the entry of oxygen molecules and increased the oxygen resistance of the film.

### 3.13. Antibacterial Activity

The CTS/CSH/TiO_2_/Gr film with a TiO_2_:Gr ratio of 6:4 exhibited the best water resistance, oxygen resistance, and mechanical properties; therefore, it was selected for antibacterial activity testing. The antibacterial activities of the pure CTS/CSH, CTS/CSH/TiO_2_, and CTS/CSH/TiO_2_/Gr films against *E. coli* and *S. aureus* are presented in Figure 6 and Table 6. The pure CTS/CSH film inhibited the growth of *E. coli* and *S. aureus* by 74.67 ± 1.50% and 70.74 ± 3.63%, respectively. This effect was mainly due to the interaction between the positively charged amine groups of the CTS and the negatively charged bacterial cell film, which increased the film permeability to induce the leakage of cell contents and, ultimately, cell death [46]. The antibacterial activity of the pure CTS/CSH film significantly improved upon the introduction of the TiO_2_ NPs (*p* < 0.05), in accordance with the known ability of these NPs to kill various microorganisms, including gram-negative and gram-positive bacteria, fungi, protozoa, viruses, and bacteriophages. At present, the antibacterial activity of TiO_2_ is attributed to (i) the UV light-induced generation of hydroxyl (^•^OH) and superoxide (^•^OOH) free radicals and other reactive oxygen species, which play a key role in inhibiting bacterial growth by oxidizing the majority of unsaturated phospholipids in microbial cells [47], and (ii) the interactions between TiO_2_ NPs and bacteria, which lead to the dissolution or potential change of the microbial cell films and, hence, cell death [48]. Compared with the CTS/CSH/TiO_2_ film, the CTS/CSH/TiO_2_/Gr film featured a slightly reduced antibacterial activity, inhibiting the growth of *E. coli* and *S. aureus* by 90.33 ± 0.72% and 95.33 ± 1.02%, respectively. It is possible that the larger number of Gr layers increased the film thickness and thereby weakened the antibacterial effect of Gr by reducing the probability of bacterial interaction [49]. It is noteworthy that the inhibitory effect of the CTS/CSH/TiO_2_/Gr film on *E. coli* was slightly lower than that on *S. aureus*, which may be related to the different cell structures of these bacteria. Gram-positive bacteria have a cytoplasmic film and a thick wall comprising multiple peptidoglycan layers and containing numerous holes that can increase susceptibility to intracellular transduction and hence facilitate the destruction of cells by nanofillers [17]. On the other hand, gram-negative bacteria have complex cell walls with a peptidoglycan layer between the outer film and cytoplasmic films; therefore, they are less vulnerable to attack by nanofillers [3]. Specifically, a 30-min ultrasonication at 160 W increased the *E. coli* and *S. aureus* inhibition efficiencies of this film to 96.67 ± 0.09% (by 6.56%) and 99.85 ± 0.13% (by 4.53%), respectively, which was in good agreement with the results reported by Zhang et al. [16]. This phenomenon was attributed to the effective dispersion of the nanofillers into smaller aggregates at a high ultrasonication power, which increased the specific surface area of the TiO_2_ and Gr particles and the probability of contact between these nanofillers and microbial cells, thereby enhancing the antibacterial activity of the film [50]. In addition, the kinetic energy and thermal effects of ultrasonication facilitated the release of the TiO_2_ and Gr particles into the polymer structure, also increasing the antibacterial activity [41].

## 4. Conclusions

In this study, we prepared CTS/CSH/TiO_2_/Gr films via ultrasonic-assisted electrospray deposition and solution-casting methods to evaluate the effects of ultrasonication power and the TiO_2_:Gr mass ratio on the films’ structure and performance. The films strongly inhibited the proliferation of two common foodborne pathogens, namely, *E. coli* and *S. aureus*; exhibited excellent UV and visible light-shielding properties; and possessed higher water and oxygen resistances than those of the nanofiller-free films. Ultrasonication promoted the uniform distribution of nanofillers in the CTS/CSH matrix and thus improved the mechanical and antibacterial properties of the composite films. The TS and E values of the composite film with a ratio of TiO_2_:Gr = 6:4 were 109.75% and 135.25% higher than those of the CTS/CSH film, respectively, which indicated the higher durability of the former packaging material. The higher water resistance can effectively prevent water permeation into the film from the environment or film damage during application, thereby positively affecting the quality of food. The high oxygen resistance and excellent UV and visible light-shielding characteristics protect food nutrients and prevent lipid oxidation and food discoloration. Finally, the excellent antibacterial properties may effectively prevent microbial infection during food storage, inhibit the growth of microorganisms, and preserve the food quality. Therefore, the CTS/CSH/TiO_2_/Gr films have a high application potential in the field of food packaging.

## Figures and Tables

**Figure 1 polymers-14-03844-f001:**
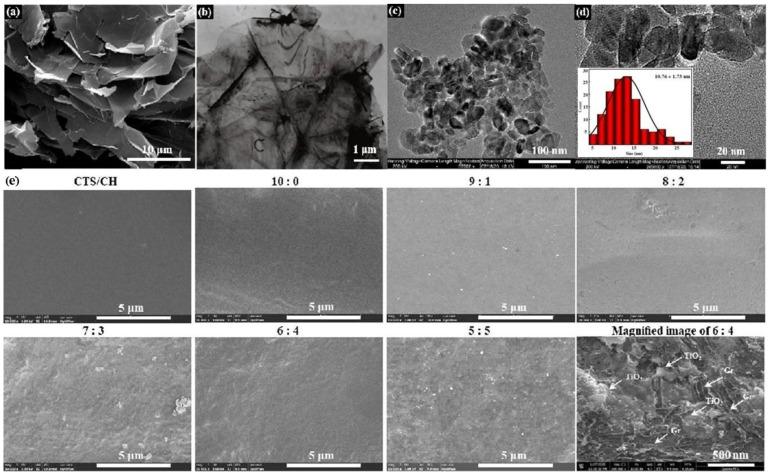
(**a**) SEM and (**b**) TEM images of Gr powder, (**c**) and (**d**) TEM images of TiO_2_ NPs, and (**e**) typical SEM images of the different composite films fabricated at an ultrasonic power of 160 W.

**Figure 2 polymers-14-03844-f002:**
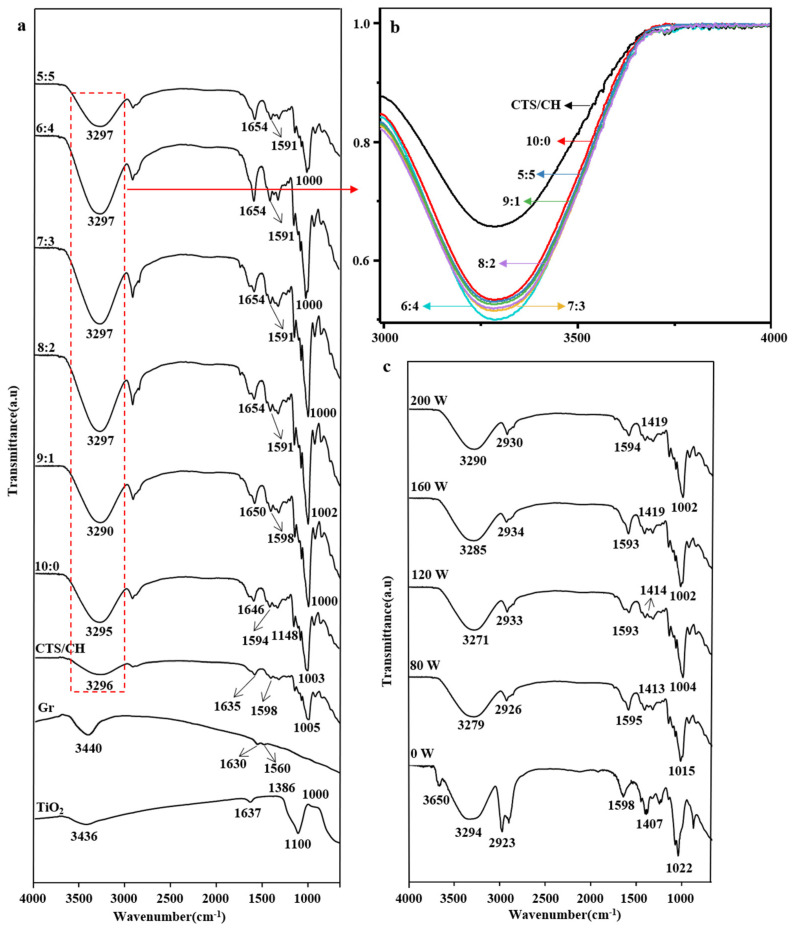
(**a**) FTIR spectra of the CTS/CSH, CTS/CSH/TiO_2_, and CTS/CSH/TiO_2_/Gr composite films; (**b**) Graph normalized to the hydroxyl peak intensity; (**c**) Typical CTS/CSH/TiO_2_/Gr composite film (TiO_2_:Gr = 6:4) fabricated at various ultrasonication powers.

**Figure 3 polymers-14-03844-f003:**
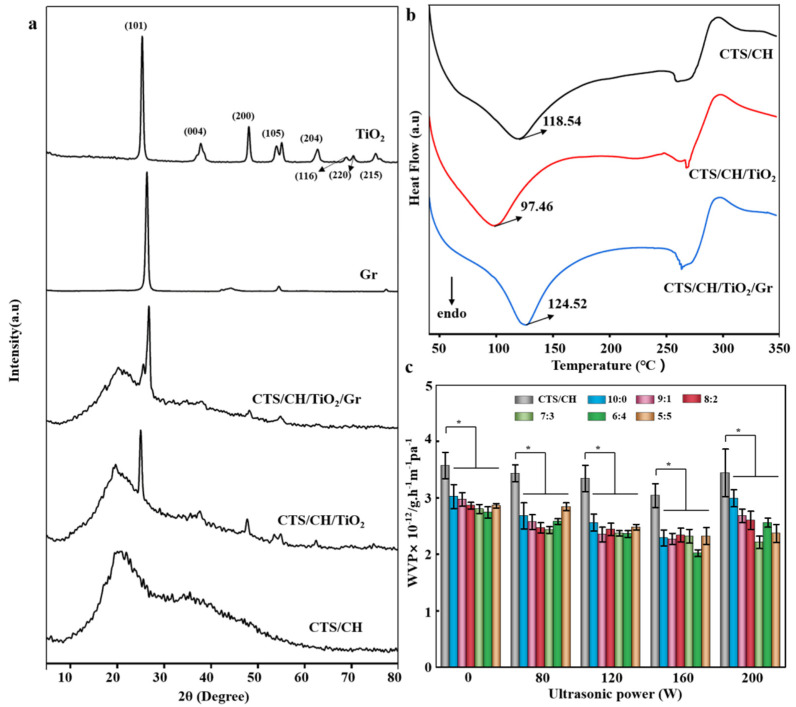
(**a**) XRD patterns of TiO_2_, Gr, and the CTS/CSH, CTS/CSH/TiO_2_, and typical CTS/CSH/TiO_2_/Gr composite films (TiO_2_:Gr = 5:5, ultrasonication power: 160 W); (**b**) DSC thermograms of the pure CTS/CSH, CTS/CSH/TiO_2_, and CTS/CSH/TiO_2_/Gr (TiO_2_: Gr = 6:4) composite films fabricated at an ultrasonication power of 160 W; (**c**) WVPs of the CTS/CSH, CTS/CSH/TiO_2_, and CTS/CSH/TiO_2_/Gr composite films fabricated at various ultrasonication powers. * *p* < 0.05.

**Figure 4 polymers-14-03844-f004:**
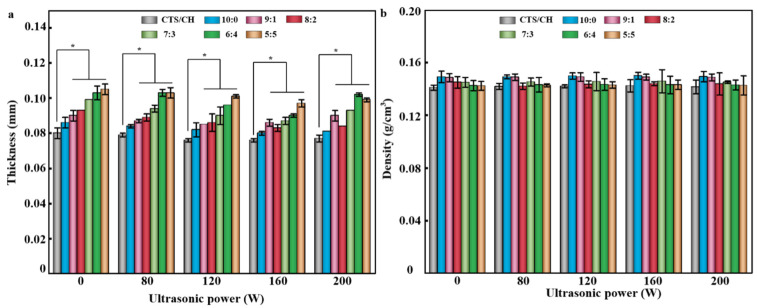
(**a**) Thicknesses and (**b**) densities of the CTS/CSH, CTS/CSH/TiO_2_ and CTS/CSH/TiO_2_/Gr composite films fabricated at various ultrasonication powers. * *p* < 0.05.

**Figure 5 polymers-14-03844-f005:**
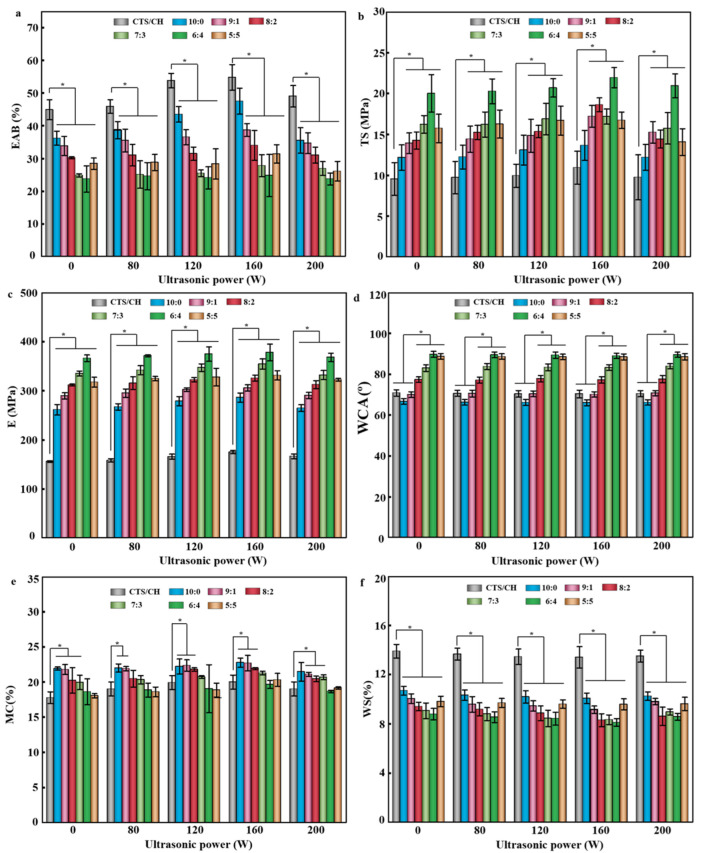
(**a**) EAB, (**b**) TS, (**c**) E, (**d**) WCA, (**e**) MC (**f**) WS values of the CTS/CSH, CTS/CSH/TiO_2_, and CTS/CSH/TiO_2_/Gr composite films fabricated at various ultrasonication powers (* *p* < 0.05, *n* = 7).

**Figure 6 polymers-14-03844-f006:**
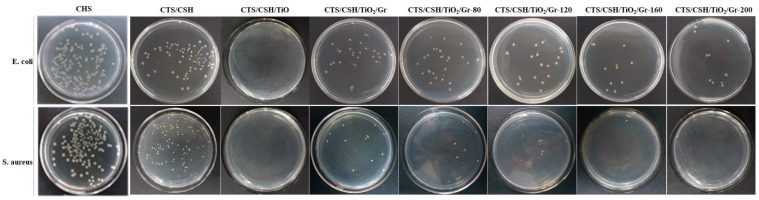
Effect of CSH, CTS/CSH, CTS/CSH/TiO_2_, and different CTS/CSH/TiO_2_/Gr composite membranes on the growth inhibition of *E. coli* and *S. aureus* for 24 h after incubation at 37 °C.

**Table 1 polymers-14-03844-t001:** Zeta potentials of the CTS/CSH, CTS/CSH/TiO_2_, and various CTS/CSH/TiO_2_/Gr composite films.

Samples	Ultrasonic Powers (W)	Zeta Potential (mV)
CTS/CSH	160	−27.33 ± 0.88 ^a^
CTS/CSH/TiO_2_	160	−14.33 ± 0.45 ^c^
9:1	160	−10.31 ± 0.13 ^de^
8:2	160	−13.27 ± 0.69 ^cd^
7:3	160	−15.27 ± 0.93 ^c^
5:5	160	−16.26 ± 0.26 ^bc^
6:4	0	−8.40 ± 0.22 ^e^
6:4	80	−12.67 ± 2.52 ^cd^
6:4	120	−15.20 ± 1.16 ^c^
6:4	160	−19.76 ± 2.28 ^b^
6:4	200	−19.73 ± 1.31 ^b^

Different letters in the same column indicate significant differences (*p* < 0.05).

**Table 2 polymers-14-03844-t002:** Thermal characteristics of CTS/CH, CTS/CH/TiO_2_, and CTS/CH/TiO_2_/Gr (TiO_2_: Gr = 6:4) composite films with ultrasonic power of 160 W.

Samples	T_o_ (°C)	T_m_ (°C)	Melting Enthalpy (∆*H_m_*, J/g)
CTS/CH	77.80	111.51	92.44
CTS/CH/TiO_2_	75.50	108.53	114.20
CTS/CH/TiO_2_/Gr	83.32	119.50	296.48

**Table 3 polymers-14-03844-t003:** Color of pure CTS/CH, CTS/CH/TiO_2_, and different CTS/CH/TiO_2_/Gr composite membranes under different ultrasonic power.

	Ultrasonic Powers (W)	Samples						
		CTS/CH	10:0	9:1	8:2	7:3	6:4	5:5
L	0	86.54 ± 0.03 Aa	87.78 ± 0.04 Ba	59.40 ± 0.35 Cabc	46.58 ± 0.83 Da	41.40 ± 0.09 Ea	43.54 ± 0.23 Fa	30.79 ± 0.01 Ga
	80	86.53 ± 0.01 Aa	87.50 ± 0.33 Aa	61.03 ± 0.32 Ba	47.05 ± 0.64 Ca	41.60 ± 0.81 Da	38.90 ± 0.93 Eb	34.48 ± 0.27 Fb
	120	86.70 ± 0.05 Aab	87.91 ± 0.01 Ba	58.27 ± 0.07 Cbc	46.08 ± 0.16 Da	41.49 ± 0.12 Ea	39.19 ± 0.04 Fb	39.73 ± 0.55 Fd
	160	87.11 ± 0.12 Ab	88.01 ± 0.01 Aa	57.97 ± 0.97 Bc	46.78 ± 0.45 Ca	41.73 ± 0.10 DEa	40.30 ± 0.09 Eb	42.73 ± 0.04 De
	200	86.76 ± 0.22 Aab	87.88 ± 0.02 Aa	59.98 ± 0.45 Bab	46.91 ± 0.83 Ca	41.42 ± 0.05 Da	30.40 ± 0.07 Ec	38.71 ± 0.05 Fc
a	0	3.34 ± 0.00 Aa	1.98 ± 0.00 Ba	1.04 ± 0.08 Cabc	0.64 ± 0.18 Da	0.66 ± 0.01 Da	1.48 ± 0.05 Ca	1.23 ± 0.01 CDa
	80	3.51 ± 0.01 Ab	2.11 ± 0.03 Bcd	1.32 ± 0.06 Cc	0.75 ± 0.15 DEa	0.65 ± 0.15 Ea	0.79 ± 0.02 DEd	1.10 ± 006 CDb
	120	3.42 ± 0.02 Ac	2.05 ± 0.01 Bb	0.81 ± 0.01 Dab	0.57 ± 0.04 Ea	0.74 ± 0.04 Da	0.87 ± 0.01 Dcd	1.23 ± 0.07 Ca
	160	3.40 ± 0.01 Ac	2.06 ± 0.01 Abc	0.70 ± 0.19 Da	0.80 ± 0.08 Da	0.72 ± 0.04 Da	1.15 ± 0.02 Cb	1.61 ± 0.01 Bc
	200	3.45 ± 0.02 Ac	2.14 ± 0.01 Ad	1.13 ± 0.09 Bbc	0.72 ± 0.15 Ca	0.67 ± 0.04 Ca	0.91 ± 0.02 BCc	1.04 ± 0.00 Bb
b	0	2.11 ± 0.12 Aa	5.41 ± 0.31 Ba	2.48 ± 0.09 Cab	1.18 ± 0.37 Da	1.11 ± 0.01 Da	1.17 ± 0.07 Da	3.01 ± 0.02 Ca
	80	2.05 ± 0.05 Aa	4.85 ± 0.21 Bab	2.78 ± 0.06 Cb	1.39 ± 0.28 Da	1.16 ± 0.35 Ea	1.77 ± 0.05 Db	2.80 ± 0.10 Cab
	120	1.94 ± 0.06 Aa	4.73 ± 0.06 Bb	2.32 ± 0.02 Cab	0.97 ± 0.07 Da	1.26 ± 0.08 Ea	1.90 ± 0.00 Fbc	3.06 ± 0.12 Ga
	160	1.92 ± 0.21 Aa	4.50 ± 0.06 Bbc	2.06 ± 0.28 CDa	1.42 ± 0.16 Ea	1.27 ± 0.05 Ea	2.41 ± 0.03 Cd	1.69 ± 0.02 DEc
	200	1.86 ± 0.04 Aa	3.96 ± 0.04 Bc	2.55 ± 0.09 Cab	1.28 ± 0.34 Ea	1.11 ± 0.10 Ea	1.97 ± 0.01 Dc	2.68 ± 0.03 Cd
ΔE	0	2.47 ± 0.32 Aa	5.97 ± 0.44 Ba	26.30 ± 0.35 Cabc	39.03 ± 0.83 Da	44.20 ± 0.09 ea	42.03 ± 0.23 Fa	45.86 ± 0.01 Ga
	80	2.33 ± 0.06 Aa	5.18 ± 0.07 Bb	24.69 ± 0.32 Ca	38.56 ± 0.64 Da	44.01 ± 0.81 ea	45.71 ± 1.01 efb	47.16 ± 0.28 Fb
	120	2.20 ± 0.08 Aa	5.24 ± 0.06 Bab	27.43 ± 0.07 Cbc	39.53 ± 0.16 Da	44.10 ± 0.12 ea	46.44 ± 0.02 fb	45.93 ± 0.54 Fac
	160	2.25 ± 0.17 Aa	5.04 ± 0.00 Bb	27.72 ± 0.96 Cc	38.84 ± 0.45 Da	43.87 ± 0.11 EFa	45.32 ± 0.09 fb	42.85 ± 0.04 Ed
	200	2.13 ± 0.02 Aa	4.56 ± 0.03 Bb	25.73 ± 0.44 Cab	38.70 ± 0.83 Da	44.18 ± 0.04 Ea	46.20 ± 0.07 fb	46.92 ± 0.05 Fbc

Values with the same letter are not statistically different, according to Duncan’s multiple range test at *p* < 0.05. a, b, c, d, e and f: mean values with the same letter in the same column are not significantly different (*p* > 0.05) (t = 7). A, B, C, D, E and F: mean values with the same letter in the same row are not significantly different (*p* > 0.05) (t = 7).

**Table 4 polymers-14-03844-t004:** Light transmittance values of CTS/CSH, CTS/CH/TiO_2_, and different CTS/CH/TiO_2_/Gr composite membranes under different ultrasonic powers.

Light Transmittance (%)	Ultrasonic Powers (W)		Samples					
		CTS/CSH	10:0	9:1	8:2	7:3	6:4	5:5
280 nm	0	75.64 ± 1.42 Aa	1.32 ± 0.00 Cb	1.84 ± 0.02 Ca	1.95 ± 0.24 Ca	2.32 ± 0.60 Ca	4.21 ± 0.92 Ba	4.84 ± 0.37 Ba
	80	75.24 ± 0.40 Aa	1.28 ± 0.06 Cb	1.68± 0.09 Cab	1.82 ± 0.26 Ca	2.23 ± 0.34 Ca	3.89 ± 0.20 Ba	4.50 ± 0.59 Ba
	120	74.89 ± 2.04 Aa	1.22 ± 0.02 Cbc	1.55± 0.02 Cbc	1.61 ± 0.19 Ca	2.06 ± 0.27 Ca	3.61 ± 0.43 Ba	4.47 ± 0.35 Ba
	160	74.42 ± 1.57 Aa	1.05 ± 0.10 Cc	1.42 ± 0.02 Cc	1.47 ± 0.04 Ca	1.80 ± 0.24 Ca	3.42 ± 0.58 Ba	4.37± 0.40 Ba
	200	75.98 ± 0.63 Aa	1.63 ± 0.06 Ca	1.72 ± 0.05 Cab	1.79 ± 0.12 Ca	2.63 ± 0.45 Ca	4.00 ± 0.36 Ba	4.57 ± 0.65 Ba
300 nm	0	85.97 ± 0.96 Aa	4.24 ± 0.11 Fa	9.92 ± 0.24 Ea	16.94 ± 0.09 Da	27.76 ± 1.04 Ca	34.15 ± 1.50 Ba	38.70 ± 3.63 Ba
	80	85.83 ± 1.11 Aa	3.91 ± 0.09 Fb	9.45 ± 0.20 Eab	16.45 ± 0.74 Da	27.55 ± 0.84 Ca	33.94 ± 1.62 Ba	38.35 ± 2.47 Ba
	120	85.19 ± 1.39 Aa	3.63 ± 0.05 Gc	9.30 ± 0.18 Fab	16.10 ± 0.40 Ea	27.36 ± 0.95 Da	33.21 ± 1.04 Ca	38.06 ± 1.60 Ba
	160	84.72 ± 0.50 Aa	3.54 ± 0.10 Fc	8.76 ± 0.13 Eb	15.55 ± 0.56 Da	27.32 ± 1.20 Ca	32.52 ± 0.83 Ba	37.33 ± 2.06 Ba
	200	85.42 ± 0.66 Aa	4.34 ± 0.04 Fa	9.70 ± 0.24 Ea	16.48 ± 0.83 Da	27.42 ± 1.17 Ca	33.91 ± 1.21 Ba	37.88 ± 2.30 Ba
400 nm	0	78.57 ± 2.34 Aa	0.36 ± 0.01 Ca	0.55 ± 0.03 Ca	0.57 ± 0.00 Ca	0.97 ± 0.23 Ba	1.02 ± 0.08 Ba	1.15 ± 0.05 Bab
	80	77.98 ± 0.42 Aa	0.21 ± 0.00 Cb	0.44 ± 0.06 Cab	0.57 ± 0.04 BCa	0.84 ± 0.14 Ba	0.42 ± 0.02 Cb	0.83 ± 0.20 Bb
	120	77.48 ± 0.73 Aa	0.13 ± 0.01 DEc	0.28 ± 0.07 CDEbc	0.05 ± 0.04 Ec	0.53 ± 0.08 Cab	0.35 ± 0.02 CDb	0.84 ± 0.16 Bb
	160	76.75 ± 1.74 Aa	0.03 ± 0.00 Cd	0.16 ± 0.00 Cc	0.02 ± 0.02 Cc	0.23 ± 0.03 Cc	0.13 ± 0.00 Cc	0.81 ± 0.15 Bb
	200	77.23 ± 0.29 Aa	0.25 ± 0.04 Cb	0.36 ± 0.02 Cb	0.31 ± 0.00 Cb	0.52 ± 0.03 Cab	0.41 ± 0.10 Cb	1.70 ± 0.20 Ba
600 nm	0	81.60 ± 0.70 Aa	1.42 ± 0.09 Bb	1.80 ± 0.12 Ba	1.16 ± 0.22 Ba	0.97 ± 0.11 Ba	0.95 ± 0.06 Ba	0.71 ± 0.13 Ba
	80	81.52 ± 1.55 Aa	1.27 ± 0.05 BCbc	1.57 ± 0.10 Ba	1.11 ± 0.26 Ca	0.86 ± 0.07 Ca	0.88 ± 0.09 Ca	0.32 ± 0.05 Db
	120	80.60 ± 1.38 Aa	1.20 ± 0.01 Bbc	0.98 ± 0.06 Cb	0.05 ± 0.01 Eb	0.55 ± 0.05 Db	0.71 ± 0.11 Dab	0.65 ± 0.02 Da
	160	80.12 ± 0.84 Aa	1.08 ± 0.04 Bc	0.38 ± 0.02 Dc	0.01 ± 0.00 Fb	0.25 ± 0.01 Ec	0.59 ± 0.02 Cb	0.30 ± 0.01 Eb
	200	80.71 ± 3.04 Aa	3.25 ± 0.15 Ba	0.88 ± 0.11 Cb	0.55 ± 0.05 DEb	0.77 ± 0.05 CDab	0.26 ± 0.00 Ec	0.40 ± 0.03 Eb
800 nm	0	82.27 ± 0.58 Aa	2.33 ± 0.13 Cb	3.15 ± 0.32 Ba	2.31 ± 0.31 Ca	1.63 ± 0.10 CDa	1.56 ± 0.30 CDa	1.24 ± 0.10 Da
	80	81.52 ± 0.03 Aa	2.20 ± 0.10 BCbc	1.83 ± 0.15 BCb	2.24 ± 0.18 Ba	1.58 ± 0.04 Cab	1.67 ± 0.32 BCa	0.62 ± 0.07 Db
	120	81.55 ± 1.20 Aa	1.95 ± 0.06 Bcd	1.65 ± 0.12 BCb	0.10 ± 0.01 Ec	0.92 ± 0.00 Dc	1.39 ± 0.26 Cab	0.55 ± 0.05 Db
	160	81.12 ± 0.36 Aa	1.60 ± 0.06 Bd	0.68 ± 0.07 Dc	0.02 ± 0.00 Ec	0.45 ± 0.02 Dd	1.13 ± 0.18 Cab	0.56 ± 0.02 Db
	200	81.71 ± 1.45 Aa	6.63 ± 0.21 Da	3.64 ± 0.28 Ca	1.14 ± 0.39 Eb	1.32 ± 0.14 Eb	0.52 ± 0.02 Eb	0.79 ± 0.05 Eb

Values with the same letter are not statistically different, according to Duncan’s multiple range test at *p* < 0.05. a, b, c and d: mean values with the same letter in the same column are not significantly different (*p* > 0.05) (t = 7). A, B, C, D, E and F: mean values with the same letter in the same row are not significantly different (*p* > 0.05) (t = 7).

**Table 5 polymers-14-03844-t005:** Oxygen resistances of the CTS/CSH, CTS/CSH/TiO_2_, and various CTS/CSH/TiO_2_/Gr composite films.

Samples	Ultrasonic Powers (W)	PV/g· (100 g)^−^^1^
PE	-	22.99 ± 0.24 ^b^
CTS/CSH	160	23.80 ± 0.03 ^a^
CTS/CSH/TiO_2_	160	18.79 ± 0.26 ^c^
9:1	160	18.08 ± 0.38 ^c^
8:2	160	16.98 ± 0.18 ^d^
7:3	160	16.74 ± 0.01 ^d^
5:5	160	16.55 ± 0.08 ^d^
6:4	0	16.91 ± 0.10 ^d^
6:4	80	16.73 ± 0.17 ^d^
6:4	120	16.61 ± 0.41 ^d^
6:4	160	16.48 ± 0.13 ^d^
6:4	200	16.56 ± 0.04 ^d^

Different letters in the same column indicate significant differences (*p* < 0.05).

**Table 6 polymers-14-03844-t006:** Effects of the CTS/CSH, CTS/CSH/TiO_2_, and CTS/CSH/TiO_2_/Gr composite films (TiO_2_:Gr = 6:4) on the growth inhibition rates (%) of *E. coli* and *S. aureus*.

Samples	Growth Inhibition (%)	
	*E. coli*	*S. aureus*
CSH	25.34 ± 0.20 ^d^	20.21 ± 4.61 ^b^
CTS/CSH	74.67 ± 1.50 ^d^	70.74 ± 3.63 ^b^
CTS/CSH/TiO_2_	98.50 ± 0.50 ^a^	99.80 ± 0.20 ^a^
CTS/CSH/TiO_2_/Gr	90.33 ± 0.72 ^c^	95.33 ± 1.02 ^a^
CTS/CSH/TiO_2_/Gr-80	90.67 ± 1.16 ^c^	97.59 ± 0.84 ^a^
CTS/CSH/TiO_2_/Gr-120	94.12 ± 0.35 ^b^	98.34 ± 0.66 ^a^
CTS/CSH/TiO_2_/Gr-160	96.67 ± 0.09 ^b^	99.85 ± 0.13 ^a^
CTS/CSH/TiO_2_/Gr-200	96.33 ± 0.10 ^b^	99.67 ± 0.12 ^a^

Different letters in the same column indicate significant differences (*p* < 0.05).

## Data Availability

Data will be made available on request.

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
