# Peer review of "Enhanced Antibacterial Performance of Chitosan/Corn Starch Films Containing TiO2/Graphene for Food Packaging"

_polymers, 2022, doi:10.3390/polym14183844_

Round 1
Reviewer 1 Report
In this work, Enhanced antibacterial performance of chitosan/corn starch films containing TiO2/graphene for food packaging. The idea of this research is interesting to readers. The background is well studied and the presentation of the method is very clear and sound, but there are some minor issues to be addressed:
The author should rewrite the word graphene instead of grapheme in title.
The author should provide suitable reference in the section 2.2. Preparation of CTS/CSH films.
Ref 14 & 15 should be rearrange.
In section 2.7. XRD analysis, the author should mention the wavelength range.
The author should mention the average particle size of TiO2 nanoparticles through the histogram analysis.
Author Response
In this work, enhanced antibacterial performance of chitosan/corn starch films containing TiO2/graphene for food packaging. The idea of this research is interesting to readers. The background is well studied and the presentation of the method is very clear and sound, but there are some minor issues to be addressed:
The author should rewrite the word graphene instead of grapheme in title.
Re: Done, thanks.
The author should provide suitable reference in the section 2.2. Preparation of CTS/CSH films.
Re: Added, thanks.
Ref 14 & 15 should be rearrange.
Re: Revised, thanks.
In section 2.7. XRD analysis, the author should mention the wavelength range.
Re: Added, thanks.
The author should mention the average particle size of TiO2 nanoparticles through the histogram analysis.
Re: Added, thanks.
Reviewer 2 Report
Z. Liu and X. Liu fabricated antibacterial films based on chitosan and corn starch. This manuscript has interesting results and worth publishing in Polymers journals. However, many major issues need to be resolved before final publications. My major comments are as follows.
In the title first character of each word should be capitalized. Check spelling error of graphene.
Line 10-13, Please simply this sentence and rewrite it clearly.
Line 35, it should be ren et al. check it. Line 46, 68 same problems, check these types of errors throughout the manuscript.
In the introduction section (at last) please make a separate paragraph for the main aim of this paper. Updated the recent literature (2021, 2022). Insert it https://doi.org/10.1016/j.lwt.2022.113665, https://doi.org/10.1016/j.ijbiomac.2019.05.151, Food Packaging and Shelf Life 33 (2022) 100904, 10.1016/j.ijbiomac.2018.06.126 and Materials Letters 316 (2022) 132046.
Line 100, Remove dot mark from wt.%. check these types of errors throughout the manuscript.
In section 2.2, why do authors take specific weight conc. of CTS and CSH?
Section 2.11 which standard authors follow for the mechanical test. Please mention it.
In the antibacterial test, authors should mention CFUs, MIC and IC50 information.
Authors optimized 160 W nanoparticles are uniform dispersed in polymer matrix. Therefore, authors provided other sonication power SEM and TEM images. AT least two power images for proofing aggregation type and behavior of nanoparticles. Authors can put this information in supplementary or main text.
Check superscript and subscript throughout the manuscript ex: Line 277 and 278, TiO2.
Section 3.2, where are SEM images of the films.
Line 304-306 need citations. Insert it International Journal of Biological Macromolecules 171 (2021) 457-464.
Insert subsection levels (a, b and c) inside the figure.
Line 344, need citations Cellulose 29 (2022) 2399-2411
Please calculate the crystallinity % for both XRD and DSC curves and compare. Please provide these data in tabular format. Calculate as per the literatures Journal of Luminescence 228 (2020) 117593 for XRD and insert these references and for DSC you can get these data from software.
Figures are so tightly packed please make it free for good looking and improve the resolution.
Check all tabular data. Some data is misplaced. Please provide UV full scan spectra.
In the antibacterial section which one is control, where is CFUs, MIC, and IC50 information? Authors should provide it in the manuscript.
Where is the contact angle information? Please provide its digital images with value.
Author Response
- Liu and X. Liu fabricated antibacterial films based on chitosan and corn starch. This manuscript has interesting results and worth publishing in Polymers journals. However, many major issues need to be resolved before final publications. My major comments are as follows.
In the title first character of each word should be capitalized. Check spelling error of graphene.
Re: Revised, thanks.
Line 10-13, Please simply this sentence and rewrite it clearly.
Re: We corrected the wrong description, thanks.
Line 35, it should be ren et al. check it. Line 46, 68 same problems, check these types of errors throughout the manuscript.
Re: Corrected, thanks.
In the introduction section (at last) please make a separate paragraph for the main aim of this paper. Updated the recent literature (2021, 2022). Insert it https://doi.org/10.1016/j.lwt.2022.113665, https://doi.org/10.1016/j.ijbiomac.2019.05.151, Food Packaging and Shelf Life 33 (2022) 100904, 10.1016/j.ijbiomac.2018.06.126 and Materials Letters 316 (2022) 132046.
Re: Updated, thanks.
Line 100, Remove dot mark from wt.%. check these types of errors throughout the manuscript.
Re: Removed, thanks.
In section 2.2, why do authors take specific weight conc. of CTS and CSH?
Re: Different specific weight conc. will affect the subsequent mechanical and antimicrobial properties, so the mass ratio is taken as the variable. Thanks.
Section 2.11 which standard authors follow for the mechanical test. Please mention it.
Re: ASTM was added, thanks.
In the antibacterial test, authors should mention CFUs, MIC and IC50 information.
Re: Section 2.15 was rewritten, thanks.
Authors optimized 160 W nanoparticles are uniform dispersed in polymer matrix. Therefore, authors provided other sonication power SEM and TEM images. AT least two power images for proofing aggregation type and behavior of nanoparticles. Authors can put this information in supplementary or main text.
Re: We added the SEM images of 120 W nanoparticles in polymer matrix, please check Fig. S1 in supplementary. Thanks.
Check superscript and subscript throughout the manuscript ex: Line 277 and 278, TiO2.
Re: Corrected, thanks.
Section 3.2, where are SEM images of the films.
Re: Fig. 1 was revised, thanks.
Line 304-306 need citations. Insert it International Journal of Biological Macromolecules 171 (2021) 457-464.
Re: Added, thanks.
Insert subsection levels (a, b and c) inside the figure.
Re: Done, thanks.
Line 344, need citations Cellulose 29 (2022) 2399-2411
Re: Cited, thanks.
Please calculate the crystallinity % for both XRD and DSC curves and compare. Please provide these data in tabular format. Calculate as per the literatures Journal of Luminescence 228 (2020) 117593 for XRD and insert these references and for DSC you can get these data from software.
Re: First of all, we are very sorry that we did not find the formula to calculate the crystallinity in the article you provided. Secondly, the paper focuses on whether adding some substances affects the crystallinity of the film, without considering the values of crystallinity. In addition, the data and workload of the article are very sufficient, so we did not try to calculate the crystallinity. The table of thermal characteristics of CTS/CH, CTS/CH/TiO2 and CTS/CH/TiO2/Gr (TiO2: Gr = 6:4) composite films with ultrasonic power of 160 W was added in table 2. Thanks.
Figures are so tightly packed please make it free for good looking and improve the resolution.
Re: We replaced some figures, thanks.
Check all tabular data. Some data is misplaced. Please provide UV full scan spectra.
Re: We added the UV full scan spectra in supplementary, thanks.
In the antibacterial section which one is control, where is CFUs, MIC, and IC50 information? Authors should provide it in the manuscript.
Re: CSH is the control, some information was added, thanks.
Where is the contact angle information? Please provide its digital images with value.
Re: The contact angle information is in Fig. 5 (d) WCA, and we also added the typical digital images in supplementary, thanks.
Reviewer 3 Report
In the article titled “Enhanced antibacterial performance of chitosan/corn starch films containing TiO2/grapheme for food packaging”, the authors show the synthesis of the Chitosan/corn starch/TiO2/graphene films by ultrasonic-assisted electrospray deposition and solution casting methods. Futhermore, they studied the effects of the TiO2/Gr mass ratio and ultrasonication power on their morphology and mechanical, optical, thermal, barrier, and antibacterial properties. In addition, the synthesized films with TiO2:Gr ratio of 6:4 and unltrasonication power of 160 W showed a uniform distribution of the nanofillers in the CTS/CSH matrix and good enhanced mechanical, barrier and water resistance properties.
The publication of this manuscript can be recommended in Polymers if the major comments could be addressed properly. The detailed comments are below:
- The authors should add the main properties and applications shown by the TiO2 NPs in the sentence “Moreover, TiO2 NPs were found to be capable of ultraviolet (UV) light absorption and ensured the transparency of composite edible films.” (Please add the following references: Nanomaterials 2020, 10(1), 124; ACS omega 3 (9), 11270-11277; Chemical Engineering Journal 428, 131249.
- Better organize paragraphs 3.1 and 3.2 well into a single one showing the commentary of the morphology analysis of the films and the dispersion in a univocal and linear discourse. In fact, I notice some confusion in the commentary of the SEM and TEM images between the text, the caption and the order of the images shown. Authors should correct this error. In addition, authors should show better resolution images. Regarding the TEM of the TiO2 NPs the authors should show the size distribution and depict d-spacing. Show higher resolution FE-SEM images of films. Furthermore, HR-TEM images on the films should be able to distinguish the distribution of TiO2 NPs and Graphene sheets in the nanocomposite.
- I suggest the authors include a small introductory sentence about the importance and purpose of using FTIR spectroscopy (please should add the following references: Micro and Nano Technologies 2017, Pages 73-93; ChemistryOpen 2021, 10, 1033–1040.).
- In the introduction, I advise the authors to summarize in a few lines the main properties shown by graphene (I suggest adding the following notes: Journal of Industrial and Engineering Chemistry 2014, 20, 1171–1185; Nanomaterials 2020, 10, 2549; Molecules 2020, 25, 5731).
Author Response
In the article titled “Enhanced antibacterial performance of chitosan/corn starch films containing TiO2/grapheme for food packaging”, the authors show the synthesis of the Chitosan/corn starch/TiO2/graphene films by ultrasonic-assisted electrospray deposition and solution casting methods. Futhermore, they studied the effects of the TiO2/Gr mass ratio and ultrasonication power on their morphology and mechanical, optical, thermal, barrier, and antibacterial properties. In addition, the synthesized films with TiO2:Gr ratio of 6:4 and unltrasonication power of 160 W showed a uniform distribution of the nanofillers in the CTS/CSH matrix and good enhanced mechanical, barrier and water resistance properties.
The publication of this manuscript can be recommended in Polymers if the major comments could be addressed properly. The detailed comments are below:
- The authors should add the main properties and applications shown by the TiO2 NPs in the sentence “Moreover, TiO2 NPs were found to be capable of ultraviolet (UV) light absorption and ensured the transparency of composite edible films.” (Please add the following references: Nanomaterials 2020, 10(1), 124; ACS omega 3 (9), 11270-11277; Chemical Engineering Journal 428, 131249.
Re: Yes, you are right, we added, thanks.
- Better organize paragraphs 3.1 and 3.2 well into a single one showing the commentary of the morphology analysis of the films and the dispersion in a univocal and linear discourse. In fact, I notice some confusion in the commentary of the SEM and TEM images between the text, the caption and the order of the images shown. Authors should correct this error. In addition, authors should show better resolution images. Regarding the TEM of the TiO2 NPs the authors should show the size distribution and depict d-spacing. Show higher resolution FE-SEM images of films. Furthermore, HR-TEM images on the films should be able to distinguish the distribution of TiO2 NPs and Graphene sheets in the nanocomposite.
Re: We re-organized Paragraphs 3.1. Due to the addition of two kinds of nano-materials into the film substrate, it is difficult to distinguish between them, especially in images of SEM. We tried to take many pictures of the film, it also difficult to distinguish between TiO2 NPs and Gr, and I hope you can understand. Thanks.
- I suggest the authors include a small introductory sentence about the importance and purpose of using FTIR spectroscopy (please should add the following references: Micro and Nano Technologies 2017, Pages 73-93; ChemistryOpen 2021, 10, 1033–1040.).
Re: Added, thanks.
- In the introduction, I advise the authors to summarize in a few lines the main properties shown by graphene (I suggest adding the following notes: Journal of Industrial and Engineering Chemistry 2014, 20, 1171–1185; Nanomaterials 2020, 10, 2549; Molecules 2020, 25, 5731).
Re: Added, thanks.
Round 2
Reviewer 3 Report
The authors have satisfactorily revised the manuscript. Now, the manuscript can be accepted for publication on Polymers Journal.